# Effects of Different Auditory Environments on Behavior, Learning Ability, and Fearfulness in 4-Week-Old Laying Hen Chicks

**DOI:** 10.3390/ani13193022

**Published:** 2023-09-26

**Authors:** Shuai Zhao, Weiguo Cui, Guoan Yin, Haidong Wei, Jianhong Li, Jun Bao

**Affiliations:** 1College of Animal Science and Veterinary Medicine, Heilongjiang Bayi Agricultural University, Daqing 163319, China; neaushuai@outlook.com (S.Z.); dqcuiwg@163.com (W.C.); guoanyin@foxmail.com (G.Y.); 2College of Animal Science and Technology, Northeast Agricultural University, Harbin 150030, China; 3College of Life Science, Northeast Agricultural University, Harbin 150030, China; weihaidongneau@163.com

**Keywords:** auditory stimulation, laying hen chicks, behavior, learning ability, fearfulness, welfare

## Abstract

**Simple Summary:**

Environmental enrichment to improve animal welfare has been receiving growing attention. The effect of auditory stimulation as a method of environmental enrichment is unclear for the behavior and welfare of chicks. In this study, one-day-of-age chicks were exposed to different auditory environments. The behavior, learning ability, and fearfulness of the chicks were examined to explore the effect of different auditory environments on the behavior and welfare of chicks. The findings confirmed that music stimulation of 65–75 dB had positive effects in reducing fearfulness, and music and noise of 85–95 dB reduce the expression of comforting and preening in chicks, impairing their learning ability, and increasing the level of fearfulness.

**Abstract:**

Environmental enrichment can improve animal welfare. As a method of environmental enrichment, the effect of different auditory stimulations on the behavior response and welfare of laying hen chicks has yet to be investigated. Therefore, this study was aimed at exploring the impact of various auditory exposures on the behavior, learning ability, and fear response of 4-week-old laying hen chicks. A total of 600 1-day-old chicks were randomly assigned to five different groups: C (control group), LM (Mozart’s String Quartets, 65 to 75 dB), LN (recorded ventilation fans and machinery, 65 to 75 dB), HN (recorded ventilation fans and machinery, 85 to 95 dB), and HM (Mozart’s String Quartets, 85 to 95 dB). The experiment was conducted from day 1 until the end of the experiment on day 28. Groups LM and LN were exposed to music and noise stimulation ranging from 65 to 75 dB. Groups HN and HM, meanwhile, received noise and music stimulation ranging from 85 to 95 dB. The control group (C) did not receive any additional auditory stimuli. During the experimental period, continuous behavioral recordings were made of each group of chicks from day 22 to day 28. On day 21, the PAL (one-trial passive avoidance learning) task was conducted. On days 23 and 24, OF (open field) and TI (tonic immobility) tests were performed, and the levels of serum CORT (corticosterone) and DA (dopamine) were measured. The results indicated that exposure to music and noise at intensities ranging from 85 to 95 dB could reduce comforting, preening, PAL avoidance rate, the total number of steps and grid crossings of OF, and the concentration of DA in 4 WOA chicks (*p* < 0.05), increase the freezing times of OF (*p* < 0.05); 65 to 75 dB of noise stimulation could reduce preening and total number steps of OF in 4 WOA chicks (*p* < 0.05), increase the freezing times of OF (*p* < 0.05); and 65 to 75 dB of music exposure could reduce the concentration of CORT in 4 WOA chicks (*p* < 0.05). Therefore, 65 to 75 dB of music exposure could produce positive effects on chicks and showed relatively low CORT level, whereas 85 to 95 dB of music and noise exposure could reduce comforting and preening behavior, impair learning ability, and increase the fear responses of chicks.

## 1. Introduction

Environmental enrichment is a potential approach to promote animal development, aiming to increase the expression of natural animal behavior, prevent or reduce the occurrence of abnormal behavior, improve the ability of animals to cope with stress, and enhance their psychological and physiological functions [1,2]. However, such enrichment must be economical and feasible [3], which is paramount for commercial animal farming. Such enrichment should not have any adverse effects on animals, such as increasing the probability of injury or disease [3] or reducing feed intake [4], which must be minimized to maintain high production levels.

In animal husbandry, increased environmental complexity has been used as a method of environmental enrichment [5]. Sound, as a method of environmental enrichment, has gained increasing interest due to its low cost and easy management, with people drawing inspiration from music therapy used in the human field to improve the health and welfare of livestock and captive animals [6]. Research has found that music stimulation can reduce stress response [7], enhance mobility, and decrease aggression in piglets [8]. After exposure to classical music, cattle are observed to be more relaxed, with reduced vocalizations and tongue rolling, and the herd was more likely to engage in positive social interactions [9]. Similarly, research on poultry has found that exposure to music can effectively reduce the calling sounds of chicks when they are briefly separated from their peers [10]. Another study on broiler chickens has shown that long-term music stimulation boosts live weight gain and reduces blood corticosterone [11], leading to fewer anxious behaviors [12].

In addition, environmental enrichment has the potential to promote animal cognitive development, induce positive emotional states [13], effectively improve animal resource utilization, and adapt to and cope with more complex farming environments [2,14]. Increasingly, research has confirmed that an enriched environment has a positive effect on animals’ completion of cognitive tasks [15]. Studies have found that music can improve animals’ cognitive abilities. For example, Mozart’s music can improve spatial performance in rats [16], can increase the expression levels of brain-derived neurotrophic factor (BDNF) and its receptor (TrkB), promotes neurogenesis in the hippocampus, and effectively strengthens spatial cognitive ability in rats [17]. Playing music continuously to embryos during the incubation period can promote the development of the embryo’s auditory system, reduce the stress response of newly hatched chicks, and improve their spatial orientation ability [18].

Environmental enrichment can reduce animals’ fear levels and alleviate stress. For example, adding enrichments to the rearing environment can reduce fear levels in 3-week-old chicks [19], alleviate anxiety in mice [20], and reduce weaning stress in piglets [21]. Studies on music have found that classical music can lower salivary cortisol levels in growing pigs [7] and alleviate anxiety and depressive behavior in rats [22].

As a potential method of environmental enrichment, auditory stimulation is increasingly being used to improve animals’ physical health and living environment, such as reducing abnormal behavior in captive animals and improving the cognitive abilities of rodents. However, there is limited research on using auditory stimulation to improve the health and welfare of laying hen chicks. Additionally, being a precocial species (relative independence and foraging ability after hatching), chicks have auditory abilities similar to adult chickens from the first day of hatching, they also possess strong learning and memory abilities and can complete various cognitive tests [2]. Therefore, it is necessary to investigate the effects of auditory stimulation on the behavior, learning ability, and fear response of chicks. In this study, we evaluated the effects of music and noise with different sound intensities on behavioral performance, learning ability, and fear levels of laying hen chicks.

## 2. Materials and Methods

### 2.1. Animals and General Procedures

All experiments were approved by and conducted according to the guidelines of the Science and Technology Ethics Committee of the Heilongjiang Bayi Agricultural University (ethics code: DWKJXY20230045).

This study used Roman-white laying hen chicks as the experimental animals. A total of 600 1-day-old chicks weighing 43 ± 4 g were purchased from a commercial hatchery and randomly divided into 5 groups, with 120 chickens per group and 8 replicates per group, with 15 chickens per replicate. The chicks were raised in 5 artificial climate rooms, with cage sizes of 98 cm × 92 cm × 45 cm and a stocking density of 600 cm^2^/chick. The experiment lasted 28 days, during which the chicks had ad libitum access to feed and water. Standard chick feed was used for feeding the chicks (crude protein 18.0%, crude fibre 7.0%, crude ash 8.0%, calcium 0.4–1.5%, total phosphorus 0.3%, sodium chloride 0.3–1.2%, water 14%, and methionine 0.4–0.9%; the main ingredients included grains and their processed products, oilseeds and their processed products, fish meal, dicalcium phosphate, stone powder, sodium chloride, amino acids, vitamins, trace elements, mold inhibitors, antioxidants, etc.). The temperature, humidity, ventilation, and lighting in the artificial climate rooms were controlled by a central controller. The temperature was set at 35–37 °C for days 1–3, 33–34 °C for days 4–7, 28–29 °C for day 8–14, 26–27 °C for days 15–21, 22–24 °C for days 22–28. During the experiment, an artificial lighting system was used, with 23 h of light and 1 h of darkness per day for days 1–3, and then a reduction of 1 h of light per day until day 14, when the lighting was set at 12 h of light and 12 h of darkness (lighting time was from 7:00 to 19:00), with a light intensity of 20 lux and a humidity of 60–70%. The chicks were immunized according to routine procedures during the experiment.

### 2.2. Experimental Groups and Auditory Stimulation

The five groups were:

Group C: Control, the chicks were reared without any additional auditory exposure, and the background sound of the control group was below 40 dB;

Group LM: The chicks were reared in the exposure to Mozart music (Mozart’s String Quartets, K428, K525, and K458 [23]), with the sound intensity set at 65 to 75 dB;

Group LN: The chicks were reared in the exposure to noise (recorded ventilation fans and machinery) with the sound intensity set at 65 to 75 dB;

Group HN: The chicks were reared in the exposure to noise (recorded ventilation fans and machinery) with the sound intensity set at 85 to 95 dB;

Group HM: The chicks were reared in the exposure to Mozart music (Mozart’s String Quartets, K428, K525, and K458) with the sound intensity set at 85 to 95 dB.

The auditory stimulation started on day 1 and continued until the end of the experiment on day 28. The experimental groups were exposed to auditory stimulation every day, and the music and noise were played from 7:00 to 19:00. Within the 12 h period of sound playback, the player cycled through the sound (one hour on/one hour off). The sound was played automatically according to a pre-set program connected to a laptop computer via two built-in speakers (SPA311, Philips (China) Investment Co., Ltd., Shanghai, China).

### 2.3. Sample Collection

#### 2.3.1. Behavioral Observation

Video cameras (DS-2CD3T10D, Hangzhou Hikvision Digital Technology Co., Ltd., Hangzhou, China) were used to record the behavior data of the chicks during the experiment. The video cameras were fixed opposite the chick cages at an angle that allowed observation of the behavior of all chicks in the cages. From days 22–28 of the experiment, behavior was observed and recorded for a total of 4 h each day, from 8:30–10:30 and 13:30–15:30 [24]. One target animal was randomly selected from each replicate and identified by spraying on different parts (neck, back, or/and tail) of the body with black dyes. After the behavior recordings were completed, the scan sampling and instantaneous recording method were used to calculate the proportion of time spent on each state behavior (feeding, standing, lying, and walking) during each observation period, with 10 s as one time unit. The data were converted into percentages of the total observation time. For event behaviors (drinking, preening, pecking, comfort behavior, and feather pecking), the focal animal sampling and continuous recording methods were used to record each behavior occurrence for each observed chick. The total number of times each behavior occurred during the observation period was recorded [24]. The specific behaviors and their definitions are shown in Table 1.

#### 2.3.2. The One-Trial Passive Avoidance Learning (PAL) Task

The PAL is a method of testing the learning ability of chicks by exploiting their instinct to peck at objects [25], as shown in Figure 1.

The test was conducted when the chicks were 21 days old. Prior to the test, the chicks were fasted for 7 h. For each experimental group, 24 chicks were randomly selected and divided into 12 pairs. The observer placed each pair in a test box (60 cm × 50 cm × 40 cm) located in an adjacent room. The test box had three paper walls and one plastic mesh wall, allowing researchers to see inside the box from a distance without observing from above. The top of the box was covered with cardboard, and a 25-watt LED light was used to illuminate the box. Each pair of chicks was marked with a black marker for differentiation during data recording.

Before the experiment training, a sheet of white A4 paper was placed on the floor of the test box, and the chicks were placed in the test box for 3 h of adaptation training without food or water. After the adaptation training, the A4 paper was removed, and three pre-training sessions were conducted. During training, a red sheet of A4 paper containing normal food was placed in the test box to attract the chicks’ attention and induce pecking behavior. The normal food was shown to the chicks for 1 min in the first and second pre-training sessions and for 2 min in the third pre-training session. There was a 5 min interval between each pre-training session. Five minutes after the last pre-training session, a training test was conducted using a blue sheet of A4 paper containing feed soaked in 99% methyl anthranilate (MeA). MeA is an organic substance with a strong aroma but an extremely bitter taste, which is not harmful to chicks. After pecking the MeA feed, the chicks showed aversive behavior such as shaking their heads, closing their eyes, and occasionally wiping their beaks on the cage floor. Fresh MeA feed was prepared daily, ensuring the feed was moistened and kept dry during pre-training. After obvious aversive behavior was observed, the blue paper and feed were removed, and a timer was started. Two hours later, discrimination ability tests were conducted twice. In the first test, normal food was shown to the chicks for 1 min, but it was placed on a paper of the same colour (blue) as the previous MeA food. After the test, the paper and food were removed. After a 5 min interval, the second test was conducted by showing normal food to the chicks for 1 min on a sheet of paper of the same colour (red) as the one used in pre-training to hold normal food. The pecking or non-pecking behavior of each target chick was recorded during each test. During the experiment, chicks associated the colour of the paper with the taste of the food and displayed avoidance behavior towards the colour of the paper associated with MeA food but not towards the colour of the paper associated with normal food. After the test, the experimental chicks were returned to their cages.

The chicks’ learning ability was assessed by calculating the avoidance score as a per-centage (the number of chicks in each group that pecked at the red paper but not the blue paper with feed/the total number of chicks that participated in the experiment × 100%) [26]. Each chick participated in training and discrimination tests only once. Throughout the PAL test, any chick that did not peck during any of the three pre-training sessions, training test, or memory detection test was excluded from statistical analysis.

#### 2.3.3. Open Field (OF) Test

At 23 days of age, 8 chicks were randomly selected from each group and OF tested individually in a separate room. The OF test used a 1.5 m × 1.5 m square wooden board as the OF area, surrounded by 1.5 m high wooden boards. The OF area was divided into 25 square areas of 0.3 m × 0.3 m using white grid paper, as shown in Figure 2. During the test, an observer placed a chick in the central square area of the OF and covered it with a 20 cm × 20 cm × 30 cm cardboard box. After the observer left the OF, the box was lifted, and the chick’s behavior was recorded for the following 10 min using a video camera (Sony 10P, Tokyo, Japan) placed above the OF. The behaviors were recorded: freezing (the time it took for the chick to stand up after the box was lifted), total vocalizations (the total number of times the chick vocalized in the open field within 10 min), the total number of steps taken (the total number of steps taken by the chick in the open field), the number of excrements (the total number of times the chick excreted in the open field within 10 min), and the number of times a square was entered (the total number of times the chick crossed a square) [27,28].

#### 2.3.4. Tonic Immobility (TI) Test

On day 24 of the experiment, 24 chicks (3 per replicate) were randomly selected from each group for the TI test. The TI test was conducted following the methods described in the literature [28,29]. After being held for a few seconds, each chick was placed lying on its back in a U-shaped wooden groove. Then, one hand gently pressed the chick’s chest, while the other hand gently pressed its head and neck to induce TI for 15 s. When the tester slowly released the hands, the time taken for the chick to right itself and stand up was recorded as the duration of TI (no other movement except breathing and slight tremors). The tester then left the hen’s sight and observed its behavior. If the chick righted itself and stood up in less than 10 s, the induction was considered a failure, and re-induction was necessary up to three times. If the chick did not right itself and stand up within 15 min of the test, the duration of TI was recorded as 900 s.

#### 2.3.5. Sample Collection and Measure of Serum Corticosterone (CORT) and Dopamine (DA) Concentration

At 25 days of age, 8 chicks were randomly selected from each group and euthanized for blood collection. To minimize stress, the capture and sampling of each chick had to be completed within 2 min. Blood samples were collected in 10 mL centrifuge tubes and allowed to clot at 4 °C for 24 h. The serum was then separated using a centrifuge (ALLSHENG iCen24, Hangzhou Aosheng Co., Ltd., Hangzhou, China) at 4000 rpm for 15 min and transferred to new 1.5 mL EP tubes and were stored at −80 °C for subsequent analysis. The serum CORT and DA level were determined using commercially available high-sensitivity ELISA kit (Nanjing Jiancheng Bioengineering Institute, Nanjing, China). The kit had a detection range from 10.2 to 60 ng/mL and demonstrated an intra-assay coefficient of variation of less than 12%.

### 2.4. Statistical Analysis

The experimental data were analyzed using SPSS 21.0 (SPSS Inc., Chicago, IL, USA). All data were assessed using the Shapiro–Wilk test to check for normal distribution, behavioral data that did not follow a normal distribution, and log transformation, and square root transformations were used to normalize the data. All data were found to follow a normal distribution. One-way ANOVA and Duncan’s multiple comparison tests were used to analyze the differences in behavior, OF test indicators, TI duration, and CORT and DA concentrations among the different groups. In the PAL test, 29% of the laying hens did not meet the test criteria and were excluded from the data analysis. The percent avoidance score in the PAL test was analyzed using the Fisher’s exact test due to the small sample size. All results were plotted using GraphPad Prism7.0, and data were presented as mean ± SEM. Differences were considered statistically significant at *p* < 0.05.

## 3. Results

### 3.1. Effects of Different Auditory Stimuli on the Behavior of 4 WOA Chicks

The effects of different auditory stimuli on the behavior of 4 WOA chicks are presented in Table 2. The HM and HN groups had a lower frequency of preening behavior than that in the C, LM, and LN groups (*p* < 0.05), while the LN group was significantly lower than the C and LM groups (*p* < 0.05), and there was no significant difference between the C and LM groups (*p* > 0.05). The comfortable behavior in the HM and HN groups was significantly lower than that in the C, LM, and LN groups (*p* < 0.05), and there was no significant difference between the HM and HN groups (*p* > 0.05) or between the LM, LN, and C groups (*p* > 0.05). Compared with the C group, no significant differences were observed in other behaviors among the groups (*p* > 0.05).

### 3.2. Effect of Different Auditory Stimuli for PAL Task of 4 WOA Chicks

The percent avoidance score of the C, LM, LN, HM, and HN groups are presented in Table 3. The C and LM groups had higher PAL avoidance scores compared to the HN and HM groups, and the differences were significant (*p* < 0.05). No significant differences were observed among the other treatment groups (*p* > 0.05).

### 3.3. Effect of Different Auditory Stimuli for Fear Levels of 4 WOA Chicks

The effects of different auditory stimuli for the indicators of the OF test and duration of TI in 4 WOA chicks are presented in Table 4. The freezing in the LN, HN, and HM groups was significantly higher compared to the C and LM groups (*p* < 0.05), and there was no significant difference between the C and LM groups (*p* > 0.05). In the LM and C groups, the total number of steps was significantly higher compared to the LN, HN, and HM groups (*p* < 0.05). The LM group was significantly higher than that in the C group (*p* < 0.05), while no significant difference was observed among the LN, HN, and HM groups (*p* > 0.05). The number of times squares were entered by the LM group was significantly higher than that in the LN, HN, and HM groups (*p* < 0.05), while no significant difference was observed between the C group and the other four groups (*p* > 0.05). No significant differences were observed among the groups in total vocalizations, number of excrements, and duration of TI (*p* > 0.05).

### 3.4. Effect of Different Auditory Stimuli on the Serum CORT and DA Levels of 4 WOA Chicks

The serum CORT concentrations of each group of chicks are shown in Figure 3A. The results showed the LM group had significantly lower CORT concentrations than the other groups (*p* < 0.05), while no significant differences were observed among the C, LN, HN, and HM groups (*p* > 0.05). The serum DA concentrations of each group of chicks are shown in Figure 3B. The HN and HM groups had significantly lower DA concentrations than the C, LM, and LN groups (*p* < 0.05), while no significant difference was observed in DA concentrations between the HN and HM groups (*p* > 0.05) or among the C, LM, and LN groups (*p* > 0.05).

## 4. Discussion

Behavioral changes in animals can reflect the interaction between individuals and their environment. Therefore, behavior is an important indicator for evaluating poultry health in poultry production [30]. The expression of comforting and preening behavior in poultry represents their relaxed state [31]. The expression of comforting and preening behavior in chicks is closely related to their rearing environment, and stressful conditions can lead to a decrease in the expression of preening and comforting behavior in chicks [32]. This study found that noise at 65–75 dB, music, and noise at 85–95 dB can significantly reduce the preening behavior of chicks, while music and noise at 85–95 dB significantly reduce the comforting behavior. This may be due to the discomfort and stress caused by the noise and high-decibel music environment in this study, which affected the normal behavior expression of the chicks. Noise as a stressor can have a negative impact on animal behavior, which has been confirmed in many studies [2,7,33]. Music, as a method of environmental enrichment in livestock and poultry farming, has been considered to have a positive impact on animals in some research [34,35]. However, this study found that music at 85–95 dB led to a decrease in preening and comforting behavior expression in chicks, which may be due to the high sound intensity, because sound above 85 dB can have adverse effects on poultry [2,6].

Many studies have shown that music stimulation can improve animals’ spatial learning ability and memory [36,37]. For example, 65–75 dB Mozart music stimulation can enhance the spatial memory [17] and learning ability of rats at different developmental stages in discrimination tasks [38]. However, this study found that, compared to the control group, 65–75 dB music stimulation did not increase the PAL avoidance score of chicks, but 85–95 dB music stimulation significantly decreased the PAL avoidance score, indicating that 85–95 dB music stimulation can impair the learning ability of chicks. At the same time, many studies have found that noise exposure can damage animals’ spatial memory [39]. Exposure to 95 dB noise can inhibit hippocampal neurogenesis in mice and impair their spatial learning ability [40]. In this study, 85–95 dB noise stimulation significantly reduced the PAL avoidance score of chicks, indicating that 85–95 dB noise can impair the learning of chicks, which is consistent with previous research results. However, 65–75 dB noise did not decrease the PAL avoidance score of chicks, which may be because the stimulation intensity was not sufficient to affect the learning ability of chicks significantly. Therefore, overall, the intensity of sound in this study is the main factor affecting the learning ability of chicks.

Fear is the most important emotional state of animals and an effective indicator for evaluating the welfare of poultry [41]. Fear is often considered a response to danger and stimulates animals to produce defensive and escape behaviors [42]. However, this defensive response can cause stress, pain, injury, suffocation, and even death in livestock [43]. Therefore, the fear response is a negative emotion and welfare state. The TI test and OF test have been proven to be effective methods for detecting poultry fear in some studies [41]. In the TI test, the longer the animal remains immobile, the more fearful it is. In the OF test, the expression of some behaviors, such as the number of excrements, the number of vocalizations, and the number of times squares are entered, are related to the intensity of fear experienced by the animal [44]. In studies on the effect of auditory stimulation on TI of poultry, classical music can reduce the duration of TI in 6 WOA broilers but has no effect in 7 WOA broilers [45]. However, Dávila et al.’s study found that classical music had no effect on the duration of TI in 8 WOA chickens of several breeds [5], which is similar to this study. In this study, it was found that two sound intensities of music stimulation lasting for 4 weeks did not have a significant effect on the TI of chicks. This suggests that the effect of music on TI of chicks may be related to breed, age, and growth experience. Interestingly, two sound intensities of noise stimulation lasting for 4 weeks also did not cause differences in the duration of TI in chicks. The study found that the level of fear decreases with the duration of stimulus exposure [29], which can explain why long-term noise stimulation in this study did not cause changes in chicks’ fear levels. In the OF test, chicks exposed to 85–95 dB music and noise showed more freezing time, total number of steps, and number of times squares were entered than 65–75 dB music stimulation. This indicates that 85–95 dB of music and noise stimulation rendered chicks more fearful of new environments, and the number of total steps and the number of times squares were entered also indicates that this fear reduced their mobility. The study found that the levels of fear tested by the TI and OF tests are fundamentally different [28]. The duration of TI is considered an avoidance strategy against predators [46], while the movement and vocalization behavior exhibited by poultry in the OF test is considered a conflict between the fear of new environments (reducing mobility) and the desire to return to the flock (increasing vocalization) [47]. The OF test suggests that chicks may reduce their activity and increase freezing time, indicating that they fear new environments and will remain motionless and quiet to avoid being discovered by predators. The TI test responds to predators, so it may not be sensitive to environmental changes. This also indicates that a single fear test is not sufficient to clarify changes in fear levels.

When animals face emotional, physiological, or environmental stress, different biological response systems will be activated to cope with the potential threats caused by stress [31,35]. The hypothalamic–pituitary–adrenal (HPA) axis is usually involved in stress response [48]. CORT is the main product of the HPA axis. Therefore, measuring the level of CORT in the body has become an important indicator for studying the degree of stress experienced by livestock [49]. The concentration of CORT in the blood changes with environmental, fear, and stress changes [50]. Studies have shown that the concentration of CORT in the blood correlated positively with the level of fear in poultry [51]. Therefore, lower cortisol levels indicate lower fear levels. In this study, 65–75 dB music stimulation reduced the serum CORT concentration of chicks, suggesting that music can enrich the environment and reduce chicks’ fear levels [52]. In addition, good animal welfare should not only be about animals not having negative experiences but should also include positive emotional expressions [53]. Most physiological indicators used to evaluate animal welfare are often negative indicators and ignore some positive physiological indicators [10,54]. Studies have proposed that DA is a positive indicator that can be used to evaluate animal welfare [55]. Research has found that music exposure can significantly increase dopamine levels in the chicken brain [36]. In this study, it was found that continuous 85–95 dB music and noise stimulation can significantly reduce the concentration of DA in chick’ serum. Overall, this may be because high-decibel sound has a negative impact on the chicks.

## 5. Conclusions

In summary, music stimulation at 65–75 dB can positively affect chicks by reducing their serum CORT concentration. However, continuous music and noise stimulation at 85–95 dB reduced the expression of chicks’ comforting and preening behaviors, impaired their learning ability, increased their fear level, and had a negative impact on their welfare.

## Figures and Tables

**Figure 1 animals-13-03022-f001:**
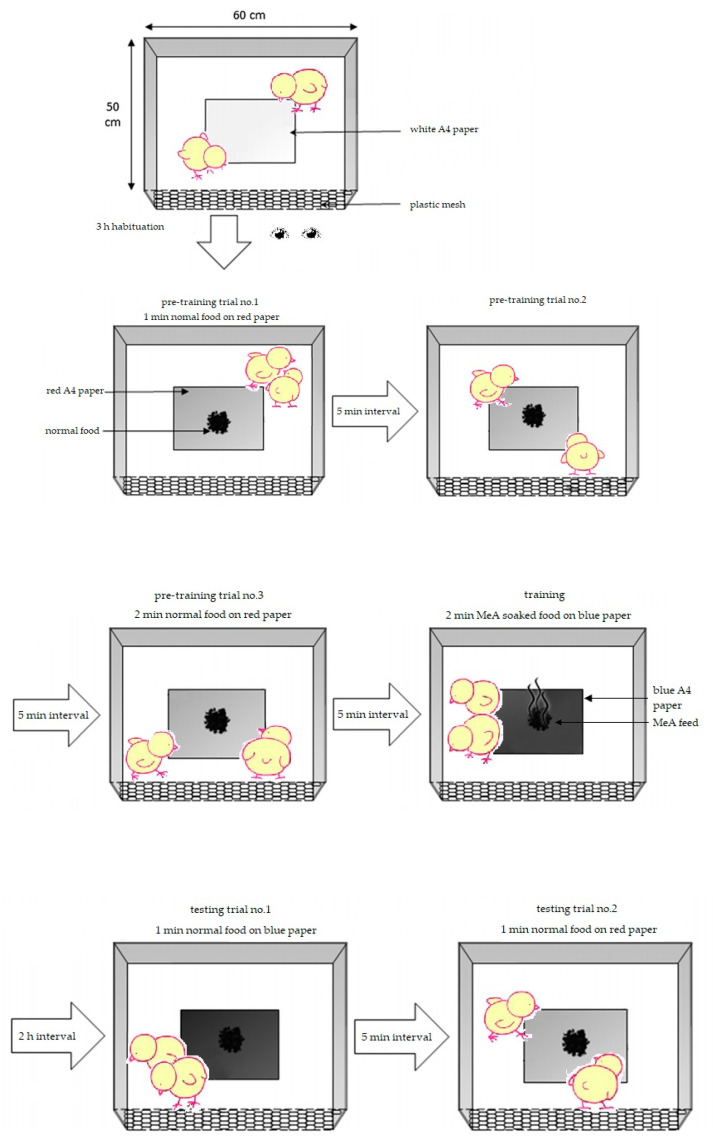
Procedural overview of the one-trail passive avoidance task. Reprinted from *Applied Animal Behaviour Science*, 207, Tahamtani, F.M.; Pedersen, I.J.; Toinon, C.; Riber, A.B., Effects of environmental complexity on fearfulness and learning ability in fast growing broiler chickens, pages 49–56, copyright (2018), with permission from Elsevier [25].

**Figure 2 animals-13-03022-f002:**
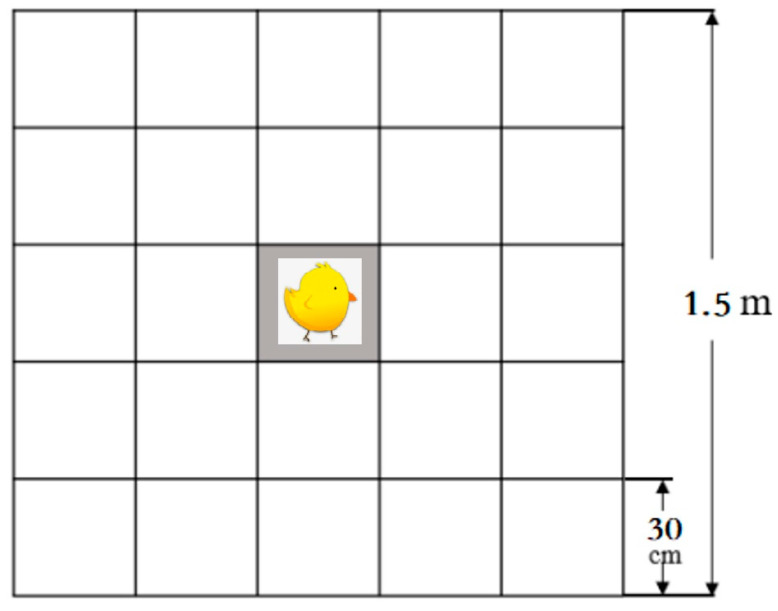
A view of open field test.

**Figure 3 animals-13-03022-f003:**
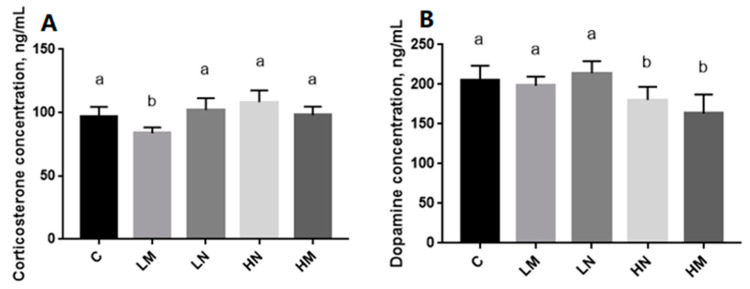
Concentration of CORT (**A**) and DA (**B**) (ng/mL) in chicks of each group exposed to auditory stimuli; ^a,b^ represent the significant difference comparison between different groups. Treatments: C (control), LM (65–75 dB music), LN (65–75 dB noise), HN (85–95 dB noise), and HM (85–95 dB music).

**Table 1 animals-13-03022-t001:** Behavior and Definition.

Behaviors	Definitions
State behavior	
Walking	Moving in one direction, including running and jumping.
Sitting	Chick sitting with its abdomen touching the ground, wings closed, and remaining inactive without any additional movements or activities.
Standing	Chick standing on its feet, with its legs extended, and displaying no movement of the body while keeping its eyes open. The head can be either in an erect position or in a relaxed posture.
Feeding	Chick directing its head towards the feed trough and engaging in pecking, ingesting feed, either once or repeatedly.
Event behavior	
Drinking	Chick approaching a nipple drinker, pecking it, and swallowing water.
Cage pecking	Chick pecks at all areas within the cage, except for the feed trough.
Preening	Chick uses its beak to gently rub, groom, and preen its feathers, or uses its toes to lightly rub its wings and head.
Comforting	Chick performs behaviors including scratching, shaking its body, wagging its tail, lifting and spreading its wings, and extending its wings and tail.
Feather pecking	Chick pecks or pulls at the feathers of other chicks, including severe feather pecking (forcefully pecking, sometimes resulting in feathers being plucked out and the pecked chicken leaving), and mild feather pecking (lightly pecking, without plucking out feathers, and the pecked chicken does not leave).

**Table 2 animals-13-03022-t002:** Effects of different sound stimuli on the behavior.

	C	LM	LN	HM	HN	SEM	F_4, 35_	*p*-Value
State behavior								
Standing (%)	29.58	30.23	28.38	29.85	28.68	1.12	0.65	0.629
Feeding (%)	32.61	31.32	32.84	34.21	31.04	2.47	0.72	0.587
Walking (%)	10.85	12.78	11.97	8.21	10.68	0.97	0.75	0.563
Sitting (%)	26.94	25.66	26.80	27.73	29.60	2.15	0.99	0.422
Event behavior								
Drinking (n)	1.08	1.21	1.11	1.34	1.29	0.33	0.98	0.429
Preening (n)	2.60 ^a^	2.71 ^a^	2.07 ^b^	1.78 ^c^	1.90 ^c^	0.19	3.34	<0.05
Comforting (n)	0.53 ^a^	0.64 ^a^	0.59 ^a^	0.39 ^b^	0.41 ^b^	0.14	2.87	<0.05
Cage pecking (n)	81.47	78.52	75.73	82.50	77.29	3.23	0.55	0.708
Feather pecking (n)	25.37	27.31	28.85	31.51	29.54	1.82	1.52	0.218

^a,b,c^ Different superscripts indicate a significant difference in different groups at *p* < 0.05. Treatments: C (control), LM (65–75 dB music), LN (65–75 dB noise), HN (85–95 dB noise), and HM (85–95 dB music).

**Table 3 animals-13-03022-t003:** Effects of different auditory stimuli on PAL task.

**Treatment**	**N**	**Avoidance (%)**	**Treatment**	** *p* ** **-Value**
C	16	62.5	LM	1.000
			LN	0.303
			HN	<0.05
			HM	<0.05
LM	16	68.75	C	1.000
			LN	0.100
			HN	<0.05
			HM	<0.05
LN	18	38.89	C	0.303
			LM	0.100
			HN	0.471
			HM	0.425
HN	17	23.53	C	<0.05
			LM	<0.05
			LN	0.471
			HM	1.000
HM	18	22.22	C	<0.05
			LM	<0.05
			LN	0.425
			HN	1.000

The chicks’ learning ability was presented for the avoidance (%). N represents the number of chicks participating in each group. The statistical difference is represented by *p* using Fisher’s exact test. Treatments: C (control), LM (65–75 dB music), LN (65–75 dB noise), HN (85–95 dB noise), and HM (85–95 dB music).

**Table 4 animals-13-03022-t004:** Effect of different auditory stimuli in open-field test and TI duration.

Measurements	C	LM	LN	HN	HM	F_4, 35_	*p*-Value
Open-field test							
Freezing (s)	164 ± 30 ^a^	124 ± 19 ^a^	270 ± 75 ^b^	247 ± 71 ^b^	255 ± 71 ^b^	4.25	<0.05
Total vocalizations (no.)	167 ± 35	189 ± 25	109 ± 24	170 ± 27	102 ± 25	1.74	0.163
Total number of steps (no.)	82 ± 29 ^b^	121 ± 31 ^a^	42 ± 12 ^c^	56 ± 17 ^c^	39 ± 11 ^c^	5.68	<0.05
Number of excrements (no.)	2.17 ± 0.31	1.67 ± 0.42	2.16 ± 0.31	2.33 ± 0.42	2.34 ± 0.33	0.82	0.521
Number of times squares entered (no.)	25.83 ± 7.35 ^ab^	42.33 ± 10.75 ^a^	16.67 ± 5.63 ^b^	12.50 ± 6.09 ^b^	14.00 ± 3.79 ^b^	6.49	<0.05
Tonic immobility test						**F_4, 115_**	** *p* ** **-value**
Duration of TI (s)	5.88 ± 1.07	5.88 ± 0.61	6.02 ± 0.43	5.71 ± 1.18	5.90 ± 0.64	0.86	0.487

Means with different superscripts ^a,b,c^ represent significant differences between groups (*p* < 0.05), and the same superscripts represent no significant differences (*p* > 0.05). Treatments: C (control), LM (65–75 dB music), LN (65–75 dB noise), HN (85–95 dB noise), and HM (85–95 dB music).

## Data Availability

Data are contained within the article.

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
