# Peer review of "Effects of Different Auditory Environments on Behavior, Learning Ability, and Fearfulness in 4-Week-Old Laying Hen Chicks"

_animals, 2023, doi:10.3390/ani13193022_

Round 1
Reviewer 1 Report (Previous Reviewer 2)
Please see attached.
Comments on animals-2610964
Title: Effects of different auditory environments on behavior, learning ability and fearfulness in 4-week-old laying hen chicks
1) This study used laying hen chicks for study targets; however, in the Methods, how can they know for sure that these 600 one-day-old chicks were all females (then laying hen chicks)? No information.
2) This MS “Effects of different auditory environments on behavior, learning ability and fearfulness in 4-week-old laying hen chicks”, was similar to Zhao et al. (2022), “Effect of prenatal different auditory environment on learning ability and fearfulness in chicks”.
It was surprising that I could not find Zhao et al. (2022) in the text and in the references.
3) This study suggested that music stimulation at 65-75dB can positively affect chicks by reducing their fear, but 85-95dB reduced the expression of chicks' comforting and preening behaviors, impaired their learning ability, increased their fear level.
Zhao et al. (2022) concluded that prenatal exposure to 65 to 75 dB music and patterned noise could not produce effective stimulation for embryonic development, but prenatal 85 to 95 dB music and patterned noise stimulation can positively reduce fearfulness in chicks.
Conclusion from this paper was some what conflict with Zhao et al. (2022), and it was difficult to image that just 10 dB difference would
Author Response
Please see the attachment

Reviewer 2 Report (Previous Reviewer 1)
Dear Authors,
I insist...Q4.L-102-what was the feed program? Starter-Grower-finisher? Was any coccidiostats use in the experiment? Did the birds get any feed additives, additives in water (vitamins, acidifiers, minerals, herbs etc.)? What was the vaccination program used in the experiment? Response4: Thank you for your comment.In this study, the feeding management procedures, immunization protocols, and disease diagnostic procedures for the chicks were implemented according to the standardized practices followed in commercial poultry farms. There is no standardisez practises in the aspects I asked- please give the details- the feeding program, feed composition, additives- all of them can affect the health, welfare and birds behaviour. The lack of this - even as a basic data is an error in methods.
Author Response
Please see the attachment.

Reviewer 3 Report (Previous Reviewer 3)
Thank you to the authors for answering my questions and doubts. I am satisfied.
The behavioral part without any reservations. The statistical part is technically correct.
In the description of the methodology, I would point out that the choice of Mozart's music was dictated by, for example, reference to literature?
From the final part of the "Discussion" chapter, I would remove the sentence suggesting that noise levels in the range of 85-95 dB inhibit dopamine secretion. This cannot be determined from the 8 individuals tested in this experiment.
In conclusions, it should be emphasized that it is not the music itself (its type), but the sound intensity that is a negative factor.
Author Response
Please see the attachment.

Reviewer 4 Report (New Reviewer)
Dear Authors,
Why was the study conducted for 4 weeks? Performance and egg quality are more important in laying hens. It would be more appropriate to include the study in these periods.
Is there a special reason for using Roman-white chickens? A sentence can be given in the introduction about this chicken genotype.
Why was sound intensity not measured in the control group?
How did you detect the chicks in camera monitoring? Considering that there are 15 animals in each subgroup, how were the chicks observed individually?
Line 36: Replace “(P>0.05)” with “(P<0.05)”
Line 38: Replace “(P>0.05)” with “(P<0.05)”
Line 113: This temperature is too high. Under normal conditions, the average environmental temperature in 28 days should be 22-24 degrees.
According to the references (27-28) given for the tonic immobility test, the test duration was taken as 600 seconds. Why did you set it as 900 seconds?
Define the abbreviations of the trial groups under the tables.
P values should be given in Table 4.
Lines 408-409: Check this sentence again. Because the 65-75dB sound intensity group did not significantly reduce fear.
Round 2
Reviewer 4 Report (New Reviewer)
Dear Authors,
The necessary explanations regarding my suggestions were found sufficient.
Best regards,
This manuscript is a resubmission of an earlier submission. The following is a list of the peer review reports and author responses from that submission.
Round 1
Reviewer 1 Report
Dear Authors,
I have revised the manuscript entitled “Effects of different auditory environments on behavior, learning ability and fearfulness in 4-week-old chicks”
This study investigated the effect of different auditory stimulation on the behaviour, learning ability, and fearfulness of 4 weeks of age (WOA) chicks.
The methodology used is classic for this type of analysis and is often used in mammal and birds` modelsThe experiment was carried out correctly, and subsequent analyzes seem to be consistent and logical. The obtained results are presented correctly, they allow to draw the conclusions presented by the authors of the work. The presented study shows a good research level, consistent with current observations in this thematic area. The graphical representation of the results as well as the general description is relatively good. The paper is interesting, but I have some other comments. Please, see below:
L-25 what does it mean-From the first day of feeding? Did the chickens not get the feed from the start of rearing, I mean did the day-old chicks get feed or not? This sentence needs to be corrected showing when the experiment started and finished (something like this: The experiment was carried out in the period from the day 0 (day old chicks` delivery and their placement in appropriate experimental groups) and the start of fattening until the 28th day of rearing. Or just like you have in L130.
L53-this sentence is incomprehensible, please correct it.
This statement is not entirely true. The subject of the influence of different types of sounds on behaviour, production parameters, etc. in poultry has been analysed for a long time. One of the first observations of this type was published in 1975 by Christensen et al. (earlier studies with tapping sounds -Tolman, 1967)
There are some nice actual studies related to sound effect on chicks` behaviour and performance that you may include in your introduction and discussion
Korsós, G., Kulcsár, M., Szabone Benyeda, Z., Glavits, R., Bersenyi, A., Gaspardy, A., & Fekete, S. G. (2019). The effect of noise and music on young meat chickens’ behaviour and stress state. Journal of Dairy, Veterinary and Animal Research, 8(3), 146-151.
Chiandetti, C., & Vallortigara, G. (2011). Chicks like consonant music. Psychological Science, 22(10), 1270-1273.
Campo, J. L., Gil, M. G., & Davila, S. G. (2005). Effects of specific noise and music stimuli on stress and fear levels of laying hens of several breeds. Applied Animal Behaviour Science, 91(1-2), 75-84.
Olczak, K., Penar, W., Nowicki, J., Magiera, A., & Klocek, C. (2023). The Role of Sound in Livestock Farming—Selected Aspects. Animals, 13(14), 2307.
Ciborowska, P., Michalczuk, M., & Bień, D. (2021). The effect of music on livestock: Cattle, poultry and pigs. Animals, 11(12), 3572.
Brouček, J. (2014). Effect of noise on performance, stress, and behaviour of animals. Slovak journal of animal science, 47(2), 111-123.
Jacob, F. G., Salgado, D. A., Nää, I. A., & Baracho, M. S. (2020). Effect of environmental enrichment on the body weight in broiler chickens. Brazilian Journal of Poultry Science, 22.
Snowdon, C. T. (2021). Animal signals, music and emotional well-being. Animals, 11(9), 2670.
L-102- what was the feed program? Starter-Grower-finisher? Was any coccidiostats use in the experiment? Did the birds get any feed additives, additives in water (vitamins, acidifiers, minerals, herbs etc.)?
What was the vaccination program used in the experiment?
What was the sex of the chicks used in the experiment?
Group LM: Low-intensity music group; the chicks were exposed to Mozart’s classical music (Mozart’s String Quartets, K428, K525, K458) stimulus of intensity ranged from 65 to 75 dB - what was the criteria for using this kind of music? Snowdon (2021) make a nice comparison of different type of music and their effect on several welfare and physiological parameters in many animal species. Mozart was one of them, but the list was longer. Moreover, violin can have high tones and may be uncomfortable for animals.
L-141- what was the criteria for choosing this age and time period? Please add the info into the manuscript.
Table 1- if it is possible- please correct the view in a way that the reader will see line to line (behaviour-definition in one line of the table.
In the discussion, you emphasize that research on behaviour has a significant impact on production. Both in the discussion and in the conclusions, there is no link to the possible use of this type of research in industrial poultry production. Please indicate how muddy (statistically significant) chicks behaviour parameters can positively affect poultry production. What additional analyses should be performed to confirm this?
After completing the above information, the study may qualify for further procedure of admission to publication.
Reviewer 2 Report
Please see attached comments.
Comments on animals-2537883
Title: Effects of different auditory environments on behavior, learning ability and fearfulness in 4-week-old chicks
This study explored effect of auditory stimulation as a method of environmental enrichment for the behavior and welfare of chicks. They showed that the music stimulation of 65-75dB could have positive effects in reducing chicks’ fearfulness; music and noise of 85-95dB reduces the expression of comforting and preening in chicks, impairing their learning ability and increasing the level of fearfulness.
Major comments:
1) they should find a gap between this study and previous work, and tell us why they should carry out this study (just copied other similar experiments, or what’s your difference); 2) mis-used references.
Minor comments:
L87-90:
However, there is a lack of systematic research on using auditory stimulation to improve the health and welfare of chicks.
This was not the case. L62, you said that research on poultry has found that exposure to music can effectively reduce the calling sounds of chicks when they are briefly separated from their peers [11].
In addition, ref. 21 was also about chicks for environmental enrichment.
L90:
4-week-old (WOA) chicks
Table 3:
Avoidance (%) then deleted all numbers’ %
Results:
All statistical analyses should give a t-, F-, or Z-value, not only showing P < 0.05 or P> 0.05.
L322-325:
You said “the study of behavior plays an important role in poultry production [34]”
But ref. 34 was about “Performance of chicks subjected to thermal challenge” and did not support your claim “the study of behavior plays an important role in poultry production”.
Also, you said “Behavior is not only an important indicator for evaluating poultry health but also the ultimate indicator for evaluating poultry adaptation to the environment [35]”
But again, ref. 35 was about “chicks exposed to low temperature”, and did not support your claim “Behavior is not only an important indicator for evaluating poultry health but also the ultimate indicator for evaluating poultry adaptation to the environment”.
Above cases showed that the authors mis-used references (just “copy or use” a sentence from a ref.), but actually the ref. was not working on what you said.
Please check all ref. cited, in particular, the Introduction and Discussion parts, to make sure that each ref. cited should be the case as you said.
L410:
5. Conclusion, not Conclusions
References:
L432:
The first reference was wrong: which journal?
In addition, for references, some journals were listed as full name, but some were in short name, indicating a poor understanding in paper writing.
Then please check all references.
Reviewer 3 Report
very interesting experience, but does it translate into practice?
It has long been known that noise has a negative impact on animal productivity.
The question is whether behavioral changes affect the productivity of livestock (in this case, chickens).
This article does not answer that question.
It is a pity that the measurement of daily weight gain was not performed.
Marking corticosteroid and dopamine - why hasn't e.g. serotonin or oxytocin been marked yet? as so-called happiness hormones.
Why was classical music chosen, and not, for example, sounds of nature: the sound of the wind, flowing stream or insects?
Article well described and documented. Readable tables. Figures significantly improve understanding of the text.
